

# An optimisation method to improve modelling of wet deposition in atmospheric transport models: applied to FLEXPART v10.4

Stijn Van Leuven[1,2,3], Pieter De Meutter[1,2], Johan Camps[1], Piet Termonia[2,3], and Andy Delcloo[2,3]

[1]Belgian Nuclear Research Centre, Mol, Belgium
[2]Royal Meteorological Institute of Belgium, Brussels, Belgium
[3]Department of Physics and Astronomy, Ghent University, Ghent, Belgium

**Correspondence:** Stijn Van Leuven (stijn.van.leuven@sckcen.be)

**Abstract.** Wet deposition plays a crucial role in the removal of aerosols from the atmosphere. Yet, large uncertainties remain in its implementation in atmospheric transport models, specifically in the parameterisation schemes that are often used. Recently, a new wet deposition scheme was introduced in FLEXPART. The input parameters for its wet deposition scheme can be altered by the user, and may be case-specific. In this paper, a new method is presented to optimise the wet scavenging rates

in atmospheric transport models such as FLEXPART. The optimisation scheme is tested in a case study of aerosol-attached
$^{137}$Cs following the Fukushima Daiichi nuclear power plant accident. From this, improved values for the wet scavenging input parameters in FLEXPART are suggested.

## 1  Introduction

Aerosols play an important role in the atmosphere, for instance through their impact on the climate and air pollution, but

also as carriers of many radionuclides such as caesium and iodine. This latter aspect is especially relevant in the context of nuclear safety. In assessing the impact of atmospheric aerosols, it is important to understand and predict their transport in the atmosphere. This can be done with state-of-the-art atmospheric transport models (hereafter ATMs).

Following an accidental or planned release of nuclear material, such as a nuclear power plant (NPP) accident or the testing and use of nuclear weapons, it is important to assess to consequences on the population and environment on both a local and

global scale. A vital tool in this context are ATMs, allowing for the prediction and simulation of the dispersion of radioactive particles in the atmosphere. On top of the dispersion there exist several removal processes, which form sinks for the aerosols. Two of these processes are dry and wet deposition. Dry deposition occurs through gravitational settling and turbulent diffusion, which can transport aerosols from the atmosphere to the Earth's surface. These particles then have to be taken up by the surface in order to be deposited. The adsorption rate depends on the terrain and type of surface cover (Seinfeld and Pandis, 2006). Wet

deposition, on the other hand, functions through scavenging processes that occur in clouds and precipitation. In general, the wet scavenging processes are described through a scavenging coefficient $\Lambda$, acting on the concentration of aerosols $c$ as

$$\frac{\mathrm{d}c}{\mathrm{d}t} = -\Lambda c. \tag{1}$$



Empirical knowledge of $\Lambda$ remains limited, resulting in large uncertainties in the modelling of wet deposition. This is exemplified by the large variety of wet deposition schemes in existing ATMs, leading to a broad range of values for the
scavenging coefficient, spanning up to four orders of magnitude (Sportisse, 2007).

Different processes contribute to the total wet scavenging, which are represented to various degrees in different ATMs. Usually, these are split up into two main categories depending on the location where the scavenging takes place: below-cloud or in-cloud. For particles in the accumulation mode (0.1-1 µm), in-cloud scavenging is said to have a larger efficiency compared to below-cloud scavenging (Baklanov and Sorensen, 2001; Andronache, 2003; Henzing et al., 2006). It is thus often thought that
in-cloud scavenging plays a dominant role in the removal of these aerosols from the atmosphere (Arnold et al., 2015; Grythe et al., 2017; Pisso et al., 2019). The difference in efficiency is a consequence of the physics driving both scavenging processes. Below-cloud scavenging is caused by falling hydrometeors (rain, snow, etc.) collecting the particulates as the hydrometeors fall towards the ground. The collections are thought to occur through various processes such as Brownian diffusion, interception and inertial impaction (Slinn, 1984; Seinfeld and Pandis, 2006; Sportisse, 2007). Theoretical wet scavenging models can take
these processes into account, but are known to underestimate the scavenging rate by 1-2 orders of magnitude for accumulation-mode particles (Wang et al., 2010, 2011). This offset can be reduced by taking into account other so-called phoretic effects (thermophoresis, diffusiophoresis and electric effects, Sportisse (2007); Jones et al. (2022)) and the rear capture effect (Jones et al., 2022). Most ATMs, however, use empirical models for scavenging, which ignore these more complex microphysical processes, and instead use fits of measured scavenging rates to certain empirical functions. These are also called parameterised
models. Most empirical studies of scavenging rates look at below-cloud scavenging. Less data exists for in-cloud scavenging, leaving its implementation in models more uncertain compared to below-cloud scavenging. In-cloud scavenging occurs through particulates acting as nuclei for the formation of cloud particles (liquid droplets or ice crystals), which then fall down as they grow larger. Accurate modelling of in-cloud scavenging is difficult, as it requires taking into account aqueous-phase chemistry.

Many sensitivity studies of wet scavenging schemes in ATMs exist (Croft et al., 2010; Draxler et al., 2015; Fang et al., 2022;
Leadbetter et al., 2015; Querel et al., 2015, 2021; Solazzo and Galmarini, 2015). These studies consist of running different ATMs with different scavenging schemes each, or running the same ATM with different scavenging schemes and altering model-parameter values. The parameter space is explored in this case by running many simulations. From these studies it is clear that no single existing ATM or deposition scheme consistently outperforms all others. The study of wet deposition thus remains an important area of research, and one where improvements to existing ATMs can make a significant impact.

In this paper, an optimisation scheme is presented which improves the modelling of wet deposition through optimising the wet scavenging coefficient. For this study, the stochastic Lagrangian particle model FLEXPART was chosen since it is a widely used state-of-the-art ATM. Furthermore, a revised wet deposition scheme was introduced in FLEXPART version 10.4 (Grythe et al., 2017; Pisso et al., 2019), which can give significantly different output compared to the old scheme when using default input parameters. Therefore in this paper we focus on the wet deposition scheme, leaving dry deposition unaltered.

In FLEXPART 10.4, the user is able to prescribe the efficiencies of below- and in-cloud scavenging. These efficiency parameters are input parameters to the simulation, and can greatly affect the resulting concentration and deposition following dispersion calculations over global scales. Determining which parameter values are appropriate is a question mark, and may





depend on a case-by-case basis (Grythe et al., 2017). Instead of exploring the parameter space of all the efficiency parameters and their possible values, we develop a method that can scale the scavenging processes post-simulation, circumnavigating the

need to explore the parameter space with individual simulations. The remaining air concentration, after scaling the scavenging processes, can then be fitted to available measurements of air concentration in order to find more appropriate values of the efficiency parameters. The proposed method is tested on the Fukushima Daiichi Nuclear Power Plant (FDNPP) accident (2011), by considering the subsequent transport and deposition of $^{137}$Cs.

The rest of this paper is structured as follows. In Sect. 2 an overview of the wet deposition scheme in FLEXPART is given.

Section 2.3 describes the methodology behind the new optimisation scheme in detail, while Sect. 3 contains the simulation setup and an overview of the observational data that was used. The results, as applied to the FDNPP case, are shown and discussed in Sect. 4.

## 2  Wet deposition of aerosols in FLEXPART 10.4

FLEXPART is a stochastic Lagrangian particle dispersion model (Stohl et al., 1998, 2005; Pisso et al., 2019), where each

released particle represents a population of aerosols or gaseous atoms. Aerosol particles are assumed to have a log-normal size distribution in FLEXPART. The removal of aerosols and gases is modelled by reducing the mass of the particles.

With FLEXPART v10.4 a new, more physically based wet deposition scheme was introduced (Grythe et al., 2017). Similar to the old scheme, the new scheme distinguishes between below - and in-cloud scavenging. With the new version, clouds are determined by the 3D cloud water fields from meteorological data. These fields are currently only provided by ECMWF. The

scavenging in FLEXPART occurs through a reduction in particle mass and takes the form of an exponential decay process during a time step $\Delta t$:

$$m(t + \Delta t) = m(t) \exp(-\Lambda \Delta t), \tag{2}$$

where $m(t)$ is the particle mass at time $t$ and $\Lambda$ the scavenging coefficient (s$^{-1}$). The previous version of FLEXPART used a simple power law of the precipitation intensity to determine the scavenging coefficients of below- and in-cloud scavenging:

$$\Lambda = AI^B, \tag{3}$$

with $I$ the precipitation intensity and $A$ and $B$ fitting parameters. The new version, as summarised in Sect. 2.1 and 2.2 below, uses a more physically based parameterisation scheme for both.

### 2.1  In-cloud scavenging

The in-cloud scavenging in FLEXPART 10.4 depends on the cloud water phase (liquid, ice or mixed) and on the nucleation

efficiency of the aerosols for serving as ice crystal or liquid droplet nuclei. In-cloud scavenging is activated inside precipitating grid cells where cloud water is also present. The precipitating cloud water (PCW) is defined as





$$\text{PCW} = \text{CTWC}\frac{F}{\text{cc}}, \tag{4}$$

where CTWC is the cloud total water content, $F$ the fraction of the grid cell experiencing precipitation and cc the surface cloud cover. $F/\text{cc}$ is then the fraction of cloud water in the precipitating part of the cloud. In-cloud scavenging occurs by aerosol particles being activated as cloud droplet or cloud ice nuclei. The quantity PCW is used in the scavenging coefficient for the in-cloud deposition scheme as

$$\Lambda = F_{\text{nuc}}\frac{I}{\text{PCW}}i_{\text{cr}}, \tag{5}$$

where $F_{\text{nuc}}$ is the nucleation efficiency (equal to the fraction of the aerosols in the cloud that are also in the cloud water), $I$ is the precipitation intensity and $i_{\text{cr}}$ is a correction factor to account for cloud water replenishment, set to a value 6.1 in FLEXPART 10.4. The nucleation efficiency $F_{\text{nuc}}$ itself is further split up into contributions from the liquid and ice water fractions ($\alpha_{\text{L}}$ and $\alpha_{\text{I}}$ respectively):

$$F_{\text{nuc}} = \alpha_{\text{L}}\text{CCN}_{\text{eff}} + \alpha_{\text{I}}\text{IN}_{\text{eff}}, \tag{6}$$

where $\text{CCN}_{\text{eff}}$ is the cloud condensation nucleation efficiency and $\text{IN}_{\text{eff}}$ the ice nucleation efficiency. For the liquid and ice water fractions, it holds that $\alpha_{\text{L}} + \alpha_{\text{I}} = 1$. The parameters $\text{CCN}_{\text{eff}}$ and $\text{IN}_{\text{eff}}$ are input parameter to the simulation, that can be changed by the user. They may depend on a case-by-case basis as there exist no unique globally representative values. The efficiencies depend on aerosol particle size, aerosol concentration and cloud properties such as updraft velocities. Furthermore, larger particles are found to have larger nucleation efficiencies compared to smaller particles. Due to the Bergeron-Findeisen process (where few ice crystals grow at the expense of many liquid droplets), it is generally assumed that for most aerosol particles $\text{CCN}_{\text{eff}} > \text{IN}_{\text{eff}}$ (Grythe et al., 2017).

## 2.2 Below-cloud scavenging

Below-cloud scavenging occurs by raindrops or snow flakes colliding, in various possible ways, with an aerosol particle and said particle staying attached to the hydrometeor. For large particle sizes, scavenging by snow is substantially more efficient than scavenging by rain. For this reason scavenging by rain and snow is calculated separately in FLEXPART. The scavenging coefficient of below-cloud scavenging by rain and snow is parameterised as

$$\log_{10}\left(\frac{\Lambda}{\Lambda_0}\right) = \log_{10}\left(C_*\right)\left(\sum_{n=0}^{4} a_n D_{\text{p}}^{-n} + b\sqrt{\frac{I}{I_0}}\right), \tag{7}$$

with $D_{\text{p}} = \log_{10}(d_{\text{p}}/d_0)$, $d_{\text{p}}$ being the mean particle diameter and $\Lambda_0 = 1\,\text{s}^{-1}$, $d_0 = 1\,\text{m}$, $I_0 = 1\,\text{mm}\,\text{h}^{-1}$ The factors $a_n$ and $b$ are fitting parameters based on observations and differ between rain (Laakso et al., 2003) and snow (Kyro et al., 2009).





The scalars $C_*$ can be adjusted by the user to change the efficiency of below-cloud scavenging by rain ($C_{\text{rain}}$) or snow ($C_{\text{snow}}$). Values from 0.1 to 10 are supposed to cover the range of below-cloud scavenging rates seen in other ATMs (Grythe et al., 2017).

Despite the fact that each particle in FLEXPART represents a population of particles with a log-normal size distribution, the size-dependent below-cloud scavenging is only calculated for the mean particle diameter. This differs from the dry deposition, which is also size dependent, but is calculated in FLEXPART by sampling the log-normal size distribution.

## 2.3 Methodology for the optimisation scheme

In order to improve the efficiency of optimising the wet deposition input parameters for FLEXPART, we propose a new systematic framework. The methodology we propose in this paper can be summarised in three steps:

1. define the contributions of individual scavenging processes for each measurement (Sect. 2.4),

2. develop an appropriate scaling scheme to the scavenging processes (Sect. 2.5),

3. minimise the error between simulations and observations (Sect. 2.6).

## 2.4 Scavenging contributions

At a given receptor, during a given time-interval, the detector will measure a concentration which is reduced due to all scavenging that has taken place in the plume on its way from the source to the receptor. We mathematically define this reduction simply as

$$c(t) = c_0(t) - \Delta c(t), \tag{8}$$

with $c_0$ the concentration if there were no scavenging and $\Delta c$ the scavenged concentration. The quantity $c_0(t)$ can be obtained through simulation by disabling all scavenging processes. Obtaining $\Delta c(t)$ is more subtle, since multiple processes can contribute:

$$\Delta c(t) = \sum_i \Delta c_i(t), \tag{9}$$

with $\Delta c_i(t)$ being the individual scavenging contribution of each process. Equations (8) and (9) can be rewritten into a single normalised summation:

$$\frac{c(t)}{c_0(t)} + \sum_i \frac{\Delta c_i(t)}{c_0(t)} = 1, \tag{10}$$





which will be useful in the further analysis. For each scavenging process, the concentration removed by scavenging can be formally noted as

$$\Delta c_i(t) = \int_{t_0}^{t} \Lambda_i(t')c(t')\mathrm{d}t', \tag{11}$$

where $t_0$ is the time of the first release at the source and $\Lambda_i$ is the scavenging coefficient of process $i$. Extracting the $\Delta c_i$'s from the FLEXPART simulations is not as straightforward as simply setting $\Lambda_i = 0$, as this method is subject to "compensation effects" which cause the contribution of the other scavenging processes to simultaneously increase. This problem arises since disabling certain scavenging processes leaves more concentration available for the other, active, scavenging processes. To avoid this problem, we directly write the $\Delta c_i$'s into the simulation output by altering the source code. This method should provide
more accurate insight into the different contributions, compared to methodologies in previous studies (e.g. Arnold et al., 2015; Grythe et al., 2017) which were done by setting $\Lambda_i = 0$.

## 2.5    Scaling scheme

The goal is to change the scavenging contributions $\Delta c_i$ in a post-process step (i.e. after obtaining the results of a simulation), in order to generate new physical air concentrations $c(t)$ with Eq. (8). Simply scaling the different contributions $\Delta c_i$ with
a scaling factor $x_i$, however, can quickly lead to non-physical negative concentrations. Therefore a more elaborate scaling scheme is proposed. With every scavenging process $i$, a scaling factor $A_i$ will be associated which acts on the concentration field through

$$\begin{aligned}
\Delta c_i(t) &= [c(t) + \Delta c_i(t)] A_i(t) \\
\\
&= \left(c_0(t) - \sum_{j \neq i} \Delta c_j(t)\right) A_i(t).
\end{aligned} \tag{12}$$

In other words, the scaling factor $A_i$ - associated with scavenging process $i$ - acts on the part of the concentration field that
is not scavenged by the other processes. Equation (12) forms a closed system of equations for all processes, which allows us to express any scavenging contribution $\Delta c_i$ as a function of all the scaling factors $A_j$ :

$$\Delta c_i(t) = c_0(t)\frac{A_i(t)}{1 - A_i(t)}\left(1 + \sum_j \frac{A_j(t)}{1 - A_j(t)}\right)^{-1}. \tag{13}$$

By this definition, the remaining concentration $c = c_0 - \sum_i \Delta c_i$ and each $\Delta c_i$ remains positive as long as every $A_i \in [0, 1[$. Here $A_i = 0$ corresponds to scavenging process $i$ not contributing at all, while in the limit $A_i \to 1$ all $c_0$ is scavenged by
process $i$. The scaling scheme of Eq. (13) also reproduces the physical compensation effect when increasing the strength of one of the scavenging processes.



## 2.6 Minimisation process

New $\Delta c_i$'s can be created by invoking a dependency of $A_i$ on a set of optimisation parameters $x_i$. As mentioned above, this dependency has to abide by $A_i(t; x_i) \in [0, 1[$. To accommodate this requirement, the following parameterisation can be chosen:

$$A_i(t; x_i) = 1 - \exp[-x_i \lambda_i(t)], \tag{14}$$

which satisfies $A_i(t; x_i) \in [0, 1[$ for $x_i \in [0, \infty[$. The values $\lambda_i(t)$ can be defined through $A_i(t, x_i = x_{i,0})$, where $x_{i,0}$ represent the reference simulation. The factors $\lambda_i(t)$ can then be interpreted to contain information about the total amount of scavenging - according to the reference simulation ($x_i = x_{i,0}$) - that has occurred inside the plume between the release and its arrival at the measurement station at time $t$. The parameterisation of Eq. (14) is chosen to resemble the exponential nature of scavenging, analogous to Eq. (2). It is worth emphasising that in defining the optimisation parameters $x_i$, a single scavenging process is not scaled independently for each measurement. Indeed, the time dependence is isolated in the immutable factors $\lambda_i(t)$. Instead $x_i$ represents a universal change in the strength of scavenging process $i$ across all times, and as such across all measurements. The values of $x_{i,0}$ can be chosen arbitrarily, and are thus set equal to 1 going forward.

Equations (8), (9), (13) and (14) can then be implemented in a numerical optimisation scheme that minimises a cost function $F$ with respect to $x_i$ such as

$$F(c, c_{\mathrm{obs}}; x_i) = \sum_k \left( \log_{10} c(t_k; x_i) - \log_{10} c_{\mathrm{obs}}(t_k) \right)^2, \tag{15}$$

where the summation is over all measurements $k$, respectively done at times $t_k$. This minimisation in logarithmic space is chosen since air-concentration measurements $c_{\mathrm{obs}}$ tend to be approximately log-normally distributed (Andersson, 2021). For the optimisation scheme, an interior-point algorithm was used to minimise the cost function Eq. (15). The numerical variation of $x_i$ was limited by demanding $x_i \in [0, 10]$. The lower bound corresponds to no scavenging by process $i$. The upper bound is arbitrarily chosen, but is not exceeded in any of the results. The initial guess of the optimisation algorithm is taken as $x_{i,0} = 1$, corresponding to the reference simulation.

## 3 Simulation setup & observational data

This section contains the information (simulation input parameters/data and observational data) needed to replicate the results shown in this paper.

### 3.1 FLEXPART input and meteodata

The numerical weather data used for this study was obtained from the MARS archive of the European Centre for Medium-Range Weather Forecast (ECMWF). It consists of 6-hourly analysis data complemented with short 3-hour forecasts and a



spatial resolution of 0.5° covering the Northern Hemisphere. It contains 91 non-uniformly spaced vertical levels, ranging from
10 m to approximately 80 km.

The methodology laid out in Sect. 2.3 is applied by using the Lagrangian particle model FLEPXART v10.4. The FLEXPART
calculations are performed with 10 million particles over the period from 11 March 2011 00:00 UTC to 5 April 2011 00:00
UTC. The results are output in 3h intervals. The $^{137}$Cs source term for the FDNPP release is taken from Terada et al. (2020) and
the $^{133}$Xe source term from Stohl et al. (2012). The relevant deposition parameters are shown in Table 1 with their initial value.
The mean particle diameter $d_\mathrm{p}$ is not the default value, but is instead chosen as to be in better agreement with measurements
from the aftermath of the FDNPP accident (Kaneyasu et al., 2012; Masson et al., 2013; Miyamoto et al., 2014). Concentration
and deposition values are output over a 3 h interval on a 0.5° grid.

| parameter | value |
|---|---|
| $C_\mathrm{rain}$ | 1 |
| $C_\mathrm{snow}$ | 1 |
| $\mathrm{CCN_{eff}}$ | 0.9 |
| $\mathrm{IN_{eff}}$ | 0.9 |
| $\rho_\mathrm{p}$ | $1900\,\mathrm{kg\,m^{-3}}$ |
| $d_\mathrm{p}$ | $0.65\,\mathrm{\mu m}$ |
| $\sigma_\mathrm{p}$ | 3 |

**Table 1.** FLEXPART v10.4 input-parameters for the deposition scheme, as used in the reference simulation of $^{137}$Cs.

### 3.2 Observational data

Measurements of $^{137}$Cs and $^{133}$Xe air concentrations are provided by the Comprehensive Nuclear-Test-Ban Treaty Organisa-
tion (CTBTO). The $^{137}$Cs data consists of 248 measurements across 20 radionuclide stations of the International Monitoring
System (IMS) that measured the highest amount of $^{137}$Cs over the simulation period (Fig. 1). For the $^{133}$Xe comparison 626
measurements were used, across the 19 IMS stations with the highest measured concentration for this radionuclide. The IMS
stations are implemented as receptors in FLEXPART, which - through the use of a parabolic kernel - give more accurate values
compared to a grid output (Stohl et al., 1998). The IMS measurements are daily averaged concentrations for $^{137}$Cs and daily
or twice-daily for $^{133}$Xe. In order to compare the FLEXPART simulations to the IMS observations, the simulation output is
averaged over the integration time of the measurements (i.e. 24 h for $^{137}$Cs and 24 h or 12 h for $^{133}$Xe).

Furthermore, wet deposition measurements of $^{137}$Cs from the National Atmospheric Deposition Program (NADP) in the
United States are used (Wetherbee et al., 2012). This data was obtained by the use of collectors that open only when it rains,
thus minimising the contamination by dry deposition. The dataset consists of measured radioactive samples from 35 NADP
sites across the contiguous United States and Alaska (Fig. 1). The deposition data of each site was integrated over a 1- to





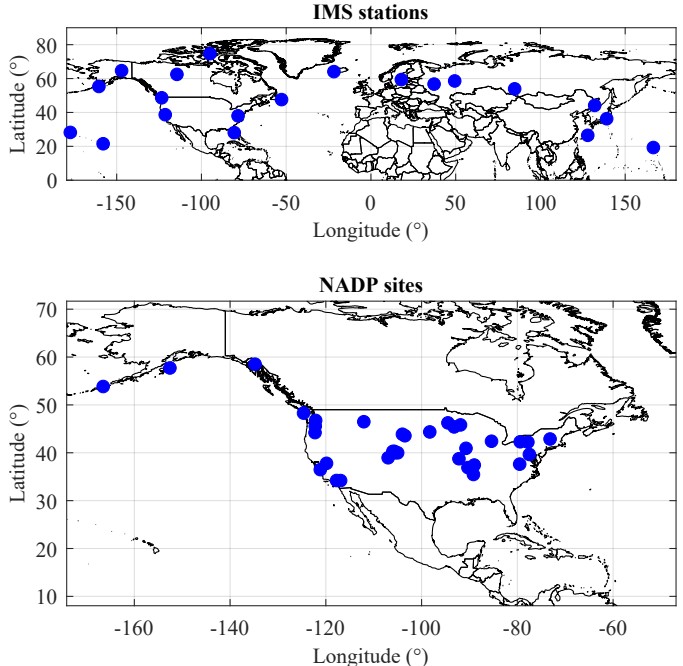

**Figure 1.** Locations of the measurement stations used in this study. Top: IMS stations. Bottom: NADP sites.

2-week period ranging from 8 March to 5 April 2011. Since FLEXPART does not allow the use of receptors for collecting deposition values, the NADP sites are compared to the closest FLEXPART grid-points.

# 4 Results & Discussion

In this section we show the results obtained by applying the above methodology to the Fukushima Nuclear accident. First,
we show the results of a FLEXPART simulation with initial parameters of Table 1, referred to as the "reference" simulation, in Sect. 4.1. Section 4.2 visualises the results of the individual scavenging contributions. The next section, Sect. 4.3, shows the results of the optimisation scheme. Finally, the results of the translation from optimisation to FLEXPART parameters are shown in Sect. 4.4.

## 4.1 Reference simulation

In order to assess whether the optimisation scheme provides an improved agreement between simulations and observations, a simulation is conducted with deposition parameters of Table 1. This simulation will be used as a reference for the sake of comparing the results of the optimisation scheme. However, of the 20 stations implemented as receptors in FLEXPART, the results of station RN38 located in Takasaki, Gunma Japan (36.3° N, 139.1° E) were troubling. The station is located only 200 km west of the FDNPP. FLEXPART produced radionuclide concentrations up to 5 orders of magnitude too low compared to



| FB | MG | NMSE | VG | R | FAC2 |
|----|----|------|----|----|------|
| (0) | (1) | (0) | (1) | (1) | (1) |
| 1.42 | 0.19 | 36.60 | 195.19 | 0.35 | 0.19 |

**Table 2.** Statistical scores of the reference simulation.

| FB | MG | NMSE | VG | R | FAC2 |
|----|----|------|----|----|------|
| (0) | (1) | (0) | (1) | (1) | (1) |
| 0.81 | 0.62 | 13.06 | 4.88 | 0.53 | 0.53 |

**Table 3.** Statistical scores of the $^{133}$Xe simulation (for $c_{\mathrm{obs}} > 10^{4.5}$ µBq m$^{-3}$).

observations, thus heavily skewing the resulting statistical analysis. Due to its proximity, it is known that the interior of the building became contaminated and showed incorrect measurements (Stohl et al., 2012). For this reason, the measurements of station RN38 were discarded in the further analysis.

Figure 2 shows the simulated air concentrations of $^{137}$Cs compared to the IMS observations, in both a scatter plot form and a histogram. With the initial deposition parameters, the simulation shows an over-estimation of the air concentration by around
a factor five. The performance of the simulation is further quantified by several statistical scores shown in Table 2. The metrics chosen are the fractional bias (FB), the geometric mean bias (MG), the normalised means square error (NMSE), geometric variance, the correlation coefficient (R) and the fraction of predictions within a factor two of observations (FAC2). A perfect model would have scores MG = VG = R = FAC2 = 1 and FB = NMSE = 0. The performance of the reference simulation is not particularly satisfying. We identify four potential sources of error that can explain this discrepancy: (i) the windfields, (ii)
the source term, (iii) the precipitation data and (iv) the deposition scheme.

In order to eliminate the possibility that the windfields cause this discrepancy, we evaluate the transport and dispersion of $^{133}$Xe during the FDNPP accident. Xe is a noble gas, and is therefore not subject to deposition. The winds are thus what drives its transport and dispersion. The scatter plot and histogram of the simulated Xe air concentrations and corresponding IMS observations are shown in Fig. 3. These show a much better agreement between observations and simulations. The observational
peak around $\sim 10^3$ µBq m$^{-3}$ is likely to originate from regulated emission sources such as medical isotope production facilities, nuclear power plants and research reactors (Gueibe et al., 2017) which were not taken into account in the simulation. Here we are only interested in the Xe originating from the FDNPP accident. Therefore, only the values above $10^{4.5}$ µBq m$^{-3}$ are considered. The statistical scores for this subset of the data are shown in Table 3. The Xe simulation performs much better than the Cs simulation. From this, it can be concluded that the winds are not mainly responsible for the large discrepancy found in
the reference $^{137}$Cs simulations.

No uncertainty range on the FDNPP source term is given by Terada et al. (2020). Looking at other source terms found in the literature, in order to make an uncertainty estimate, suggests the source term of Terada et al. (2020) is on the lower end in terms of total emitted $^{137}$Cs. The Terada et al. (2020) source term is recently revised from Katata et al. (2015), and reduced the



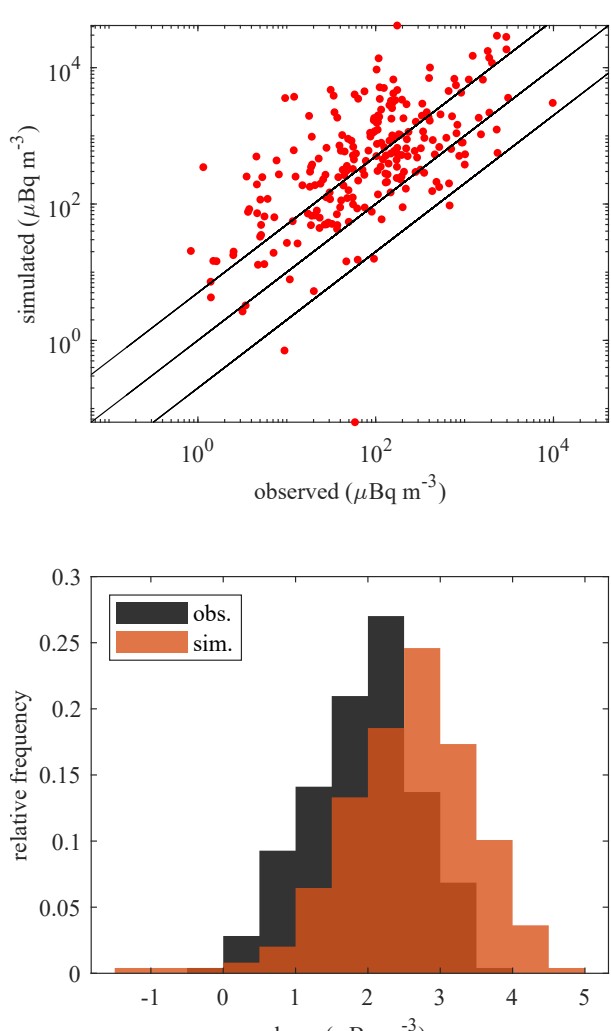

**Figure 2.** [137]Cs concentrations: reference simulation vs. IMS observations.




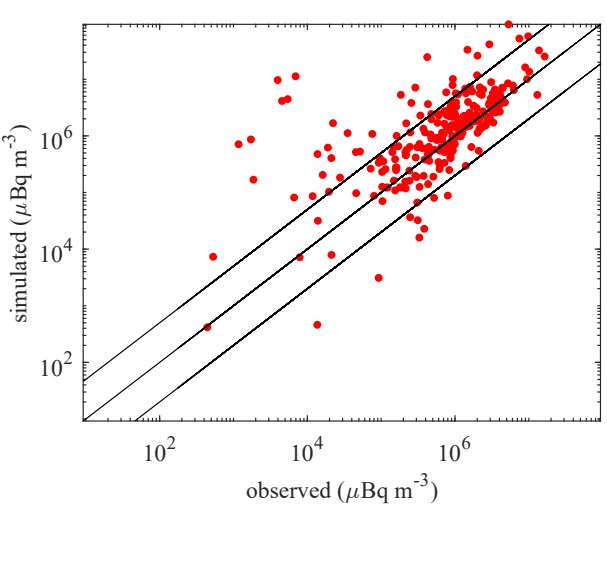

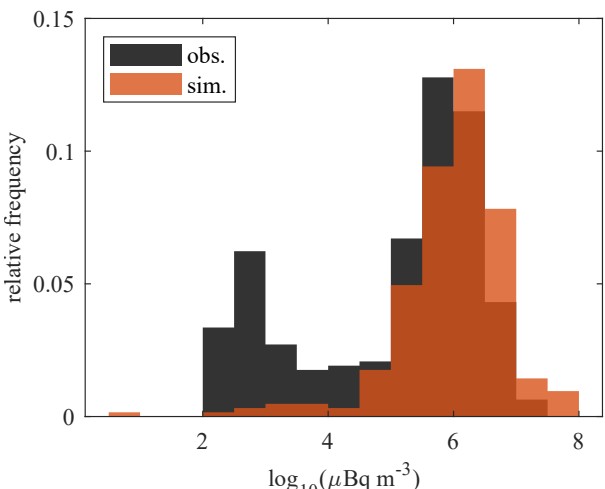

**Figure 3.** [133]Xe air concentrations: simulated vs. IMS observations.



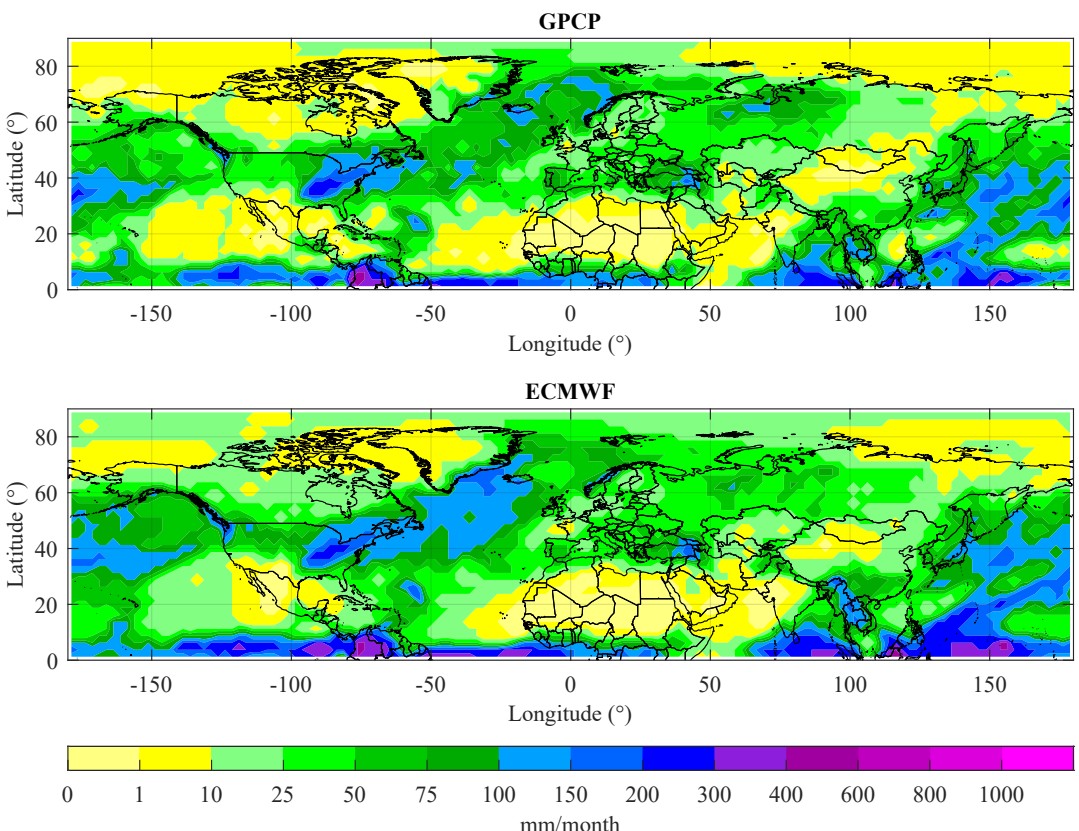

**Figure 4.** Top: GPCP precipitation data for the month of April 2011. Bottom: ECMWF integrated precipitation over the month of April 2011.

estimated total amount of emitted $^{137}$Cs by 29% from 14 PBq to 10 PBq. This is also a reduction of 73% compared to one of

the earliest source term reconstructions of the FDNPP incident by Stohl et al. (2012) which gave a total emission of 37 PBq,
with an uncertainty range of 20-53 PBq. Since FLEXPART is a linear dispersion model - meaning that the air concentrations
scale linearly with the source term - an additional reduction of a factor five in the source term would be needed to explain
the discrepancy seen in the $^{137}$Cs FLEXPART simulation. Thus, considering that the Terada et al. (2020) source term already
appears on the lower end of available estimated source terms, we consider the source term to be an unlikely main source of the

discrepancy.

   To tackle a potential flaw in the use of the precipitation data, we also compare the ECMWF precipitation (as obtained through
the use of the Flex_extract software (Tipka et al., 2020) to observations by the Global Precipitation Climatology Project (GPCP,
Adler et al., 2023, 2018). The GPCP data is based on microwave and infrared satellite measurements, and is calibrated with
rain gauge observations by the Global Precipitation Climatology Centre (GPCC, Schneider et al., 2011). Figure 4 shows the

GPCP and the ECMWF data. The precipitation is integrated over the whole month of April 2011. A good correspondence is
seen between the GPCP and the ECMWF data.



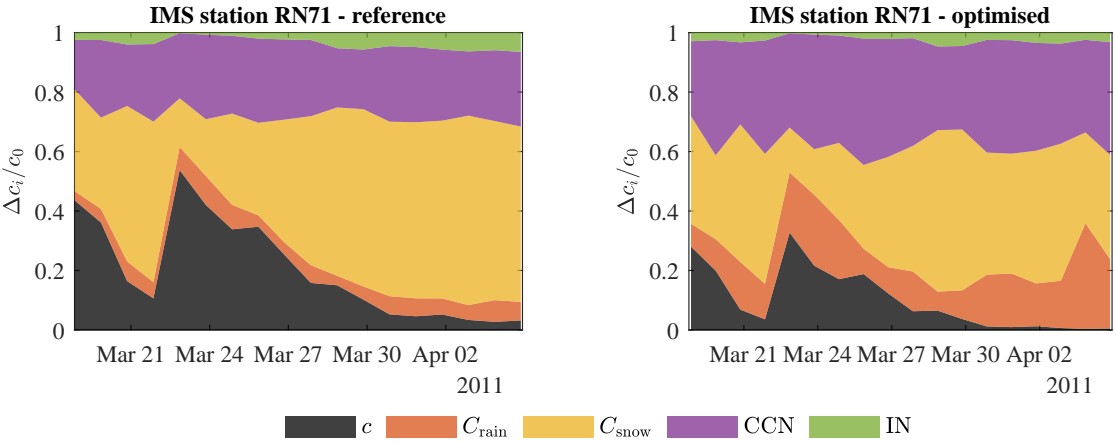

**Figure 5.** Simulated relative contributions of the different removal processes of $^{137}$Cs for the RN71 station following the FDNPP accident. Left: reference simulation. Right: optimised concentrations.

Given the above validation of the meteorological data and the source term, we conclude that the discrepancy of the reference simulation probably lies in the deposition scheme in FLEXPART. We seek an improvement by finding new, better input parameters to the wet deposition scheme compared to those in Table 1.

### 4.2 Scavenging contributions

Step one in the methodology is to quantify the individual scavenging contributions of the reference simulation (Sect. 2.4). Figure 5 (left panel) shows the scavenging contributions of the reference simulation for the IMS radionuclide station RN71 (Sand Point, Alaska US, 55.3° N 160.5° W). The horizontal axis starts on 18 March 2011, when the simulated plume first reached the receptor. The vertical axis shows the relative amount of each contribution compared to the concentration $c_0$ (i.e. the fictitious concentration that would be left over without any scavenging). The values are daily averaged concentrations as calculated by the parabolic kernel-method in FLEXPART. The relative sum of all scavenging-contributions and remaining air concentration is equal to 1, in accordance with Eq. (10). Figure 5 thus shows quantitatively which part of the concentration has been removed due to the different scavenging processes for the part of the plume that reaches the detector during a given time period.

Note that the air concentration left over after scavenging ($c$, black coloured area), as with all quantities on this plot, is relative to $c_0$. The fictitious concentration $c_0$ is determined only by the winds and not deposition, much like what would be the case with a noble gas such as Xenon. Therefore unlike the temporal variations in $c$, the variations in the ratio $c/c_0$ are not mediated by the windfields, but instead solely by spatiotemporal variations of the different removal processes. In other words: the windfield-variations are divided out by taking the quantities relative to $c_0$.





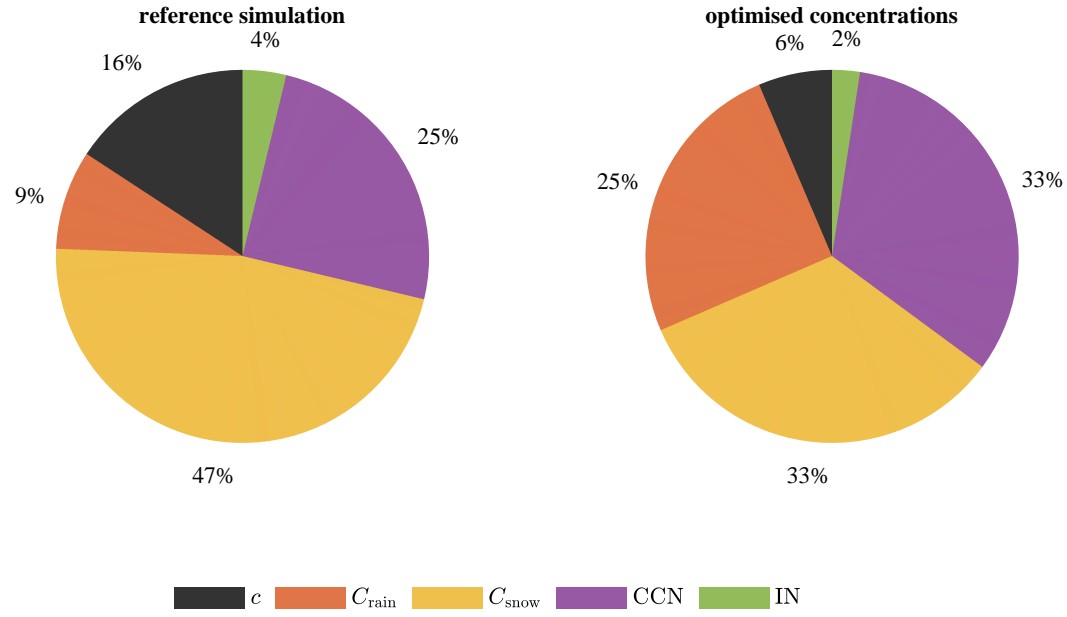

**Figure 6.** Total simulated contributions of the different removal processes over all stations. Left: reference simulation. Right: optimised concentrations.

An overview of the scavenging processes across all stations is shown in Fig. 6 (left panel). It covers the whole simulation period (11 March - 5 April 2011). In total 84% of the concentration is scavenged and deposited in the reference simulation. 66% of the wet scavenging occurs below-cloud, while 34% occurs in-cloud. The partitioning of below- and in-cloud scavenging found here is in contrast with those of some previous studies (Andronache, 2003; Henzing et al., 2006; Arnold et al., 2015; Grythe et al., 2017; Pisso et al., 2019), who found that in-cloud scavenging has a greater influence on deposition values than below-cloud scavenging. An explanation of this discrepancy can likely be found in either the use of erroneous methodology for calculating the partitioning or in the underestimation of the scavenging coefficients in the other studies. Whereas we extracted the $\Delta c_i$'s directly during a single simulation, Arnold et al. (2015), Grythe et al. (2017) and Pisso et al. (2019) performed simulation runs of the FDNPP accident with below- and in-cloud scavenging separately disabled. The latter method, however, is subject to compensation effects. Disabling a scavenging process leaves a higher air concentration for the other scavenging process(es). Indeed, by simply disabling a single scavenging process, more mass is available to the other scavenging



| FB | MG | NMSE | VG | R | FAC2 |
|----|----|------|----|----|------|
| (0) | (1) | (0) | (1) | (1) | (1) |
| 0.82 | 0.73 | 15.71 | 22.67 | 0.41 | 0.27 |

**Table 4.** Statistical scores of the optimised concentrations.

processes. This compensation effect is avoided in our methodology as the scavenging contributions are directly extracted from the simulation, through alterations of the FLEXPART source code. The methodology of Andronache (2003) and Henzing et al. (2006) is based on a theoretical approach to the below-cloud scavenging coefficient; a method which is known to underestimate the scavenging coefficient by 1-2 orders of magnitude (Wang et al., 2010, 2011). Recent measurements have indicated that

below-cloud scavenging contributes to the majority of the total wet deposition (Chatterjee et al., 2010; Xu et al., 2017; Ge et al., 2021).

### 4.3 Optimised concentrations

Quantified in the previous section, the individual removal processes were then scaled and optimised according to Sect. 2.6. The optimisation method is able to reduce the bias compared to the reference simulation, as can be seen in Fig. 7 and the statistical

scores in Table 4. An improvement in all statistical scores is seen.

The optimisation algorithm found this best fit with the $x_i$-values (Eq. (14)) shown in Table 5. All optimisation parameters have increased, suggesting the efficiency of all four scavenging processes needs increasing. The greatest increase is found for below-cloud scavenging by rain ($\times 3.6$). Below-cloud scavenging by snow is relatively little changed with a factor 1.4. In-cloud scavenging by cloud condensation nucleation (CCN) and ice nucleation (IN) are increased by slightly greater amounts ($\times 2.0$

and 1.8 respectively). The optimised scavenging contributions for station RN71 are shown in Fig. 5 (right panel). Figure 6 (right panel) shows the new removal contributions over all stations. We find that the largest contributor to the total wet deposition remains below-cloud scavenging at 63% compared to 37% for in-cloud scavenging. Compared to the reference simulations (Fig. 6, left panel), the total concentration reaching the detectors is reduced from 16% of the concentration $c_0$ to 6%. The greatest increases in scavenging are seen in collection by rain (from 9 to 25%) and cloud condensation nucleation (from 25

to 33%). One may notice that despite the increase in efficiency of collection by snow, the relative contribution thereof has decreased from 47 to 33%. This is the aforementioned compensation effect at play. A greater increase of other processes leaves less concentration available for snow collection. Similarly for ice nucleation, whose contribution has actually shrunk from 4 to 2% despite the increase in its efficiency.

### 4.4 Translation to scavenging coefficients

In order for the results of the optimisation method to be usable in future FLEXPART simulations, one has to translate the optimisation parameters $x_i$ (Table 5) to the physical scavenging coefficients which can be altered in FLEXPART through the input parameters $\{C_{\text{rain}}, C_{\text{rain}}, \text{CCN}_{\text{eff}}, \text{IN}_{\text{eff}}\}$ (Table 1). Since the methodology laid out above is only a post-process



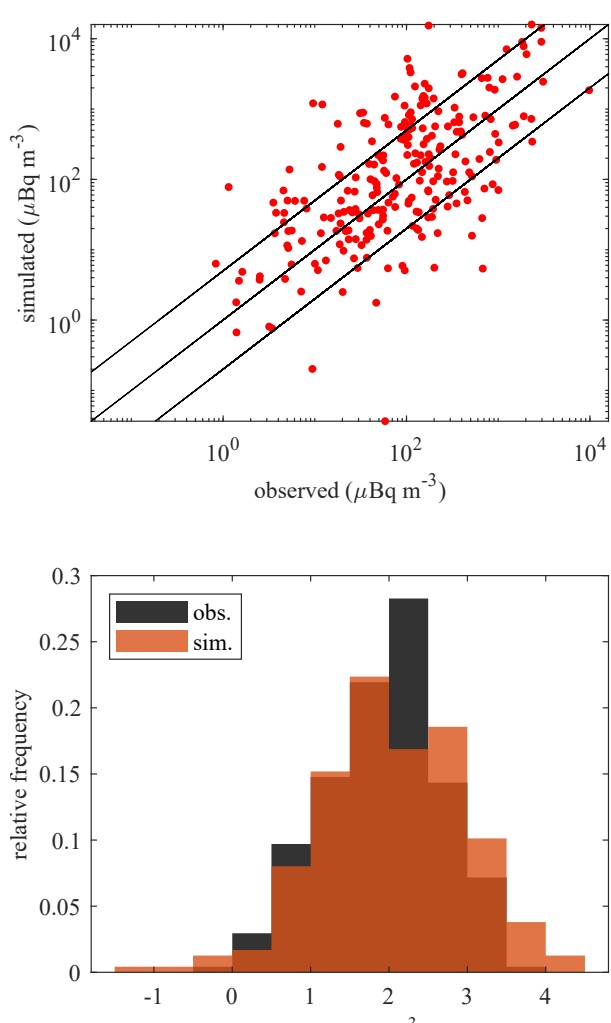

**Figure 7.** $^{137}$Cs concentrations: optimised vs. IMS observations.

|  | rain | snow | CCN | IN |
|---|---|---|---|---|
| initial ($x_{i,0}$) | 1 | 1 | 1 | 1 |
| optimised ($x_i$) | 3.6 | 1.4 | 2.0 | 1.8 |

**Table 5.** Optimisation parameters before and after the optimisation process.





| parameter | value |
|-----------|-------|
| $C_{\text{rain}}$ | 3.6 |
| $C_{\text{snow}}$ | 1.4 |
| $\text{CCN}_{\text{eff}}$ | 1.8 |
| $\text{IN}_{\text{eff}}$ | 1.6 |

**Table 6.** Optimised $^{137}$Cs wet deposition input-parameters for FLEXPART v10.4 for the FDNPP accident.

| FB | MG | NMSE | VG | R | FAC2 |
|----|----|------|----|----|------|
| (0) | (1) | (0) | (1) | (1) | (1) |
| 0.69 | 0.76 | 13.47 | 15.11 | 0.44 | 0.29 |

**Table 7.** Statistical scores of the optimised FLEXPART simulation.

approach, it is not necessarily expected to have an exact one-to-one corresponding FLEXPART simulation. Therefore the goal is to find a simulation that matches the optimised concentrations as closely as possible. We propose an ansatz that

$$\Lambda_i\Big|_{\text{optim.}} = \frac{x_i}{x_{i,0}} \times \Lambda_i\Big|_{\text{initial}}. \tag{16}$$

This a priori guess can be motivated as follows. The scaling factors $A_i$ (Eq. (14)) are defined analogous to the exponential nature of scavenging (Eq. (2)). Therefore an analogy can be drawn between $x_i \lambda_i$ and $\Lambda_i \Delta t$. By taking the fraction of optimised scavenging over initial scavenging, what remains is Eq. (16).

The new FLEXPART input parameters, according to Eq. (16), are shown in Table 6. One may notice that according to the

optimisation process, the efficiencies of in-cloud nucleation are greater than 1, which is a challenge to interpret physically. If a maximum possible efficiency of 1 is assumed, then larger values of $x_i$ can be interpreted to actually probe other factors in Eq. (5), such as the cloud water replenishment factor $i_{\text{cr}}$. Since the value of $i_{\text{cr}}$ is not an input parameter to FLEXPART, but instead defined in the source-code, one can still use the optimised values of $\text{CCN}_{\text{eff}}$ and $\text{IN}_{\text{eff}}$ as input parameters to obtain the result of increased $\Lambda$ for these processes; albeit with the caveat that one should be mindful with the physical interpretation.

Using the optimised input parameters from Table 6 in a new FLEXPART simulation leads to the $^{137}$Cs concentrations shown in Fig. 8. Already with the ansatz of Eq. (16), results are obtained that are close to the post-process optimised concentrations (Fig. 7). A comparison of the new scores in Table 7 with the post-process optimisation (Table 4) even shows a quantitative improvement for all statistical scores.

### 4.5 Independent verification with wet deposition measurements

Finally, we present an independent check of the results by looking at the wet deposition measurements from NADP. The deposition values of the reference simulation and NADP observations are shown in Fig. 9, and the corresponding scores in



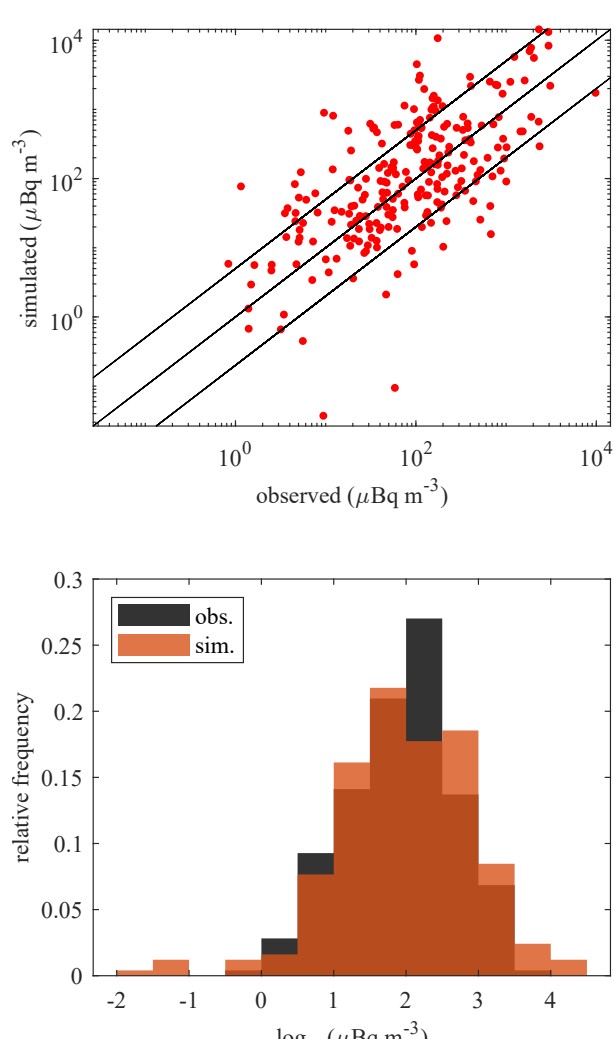

**Figure 8.** $^{137}$Cs concentrations: optimised FLEXPART simulation vs. IMS observations.



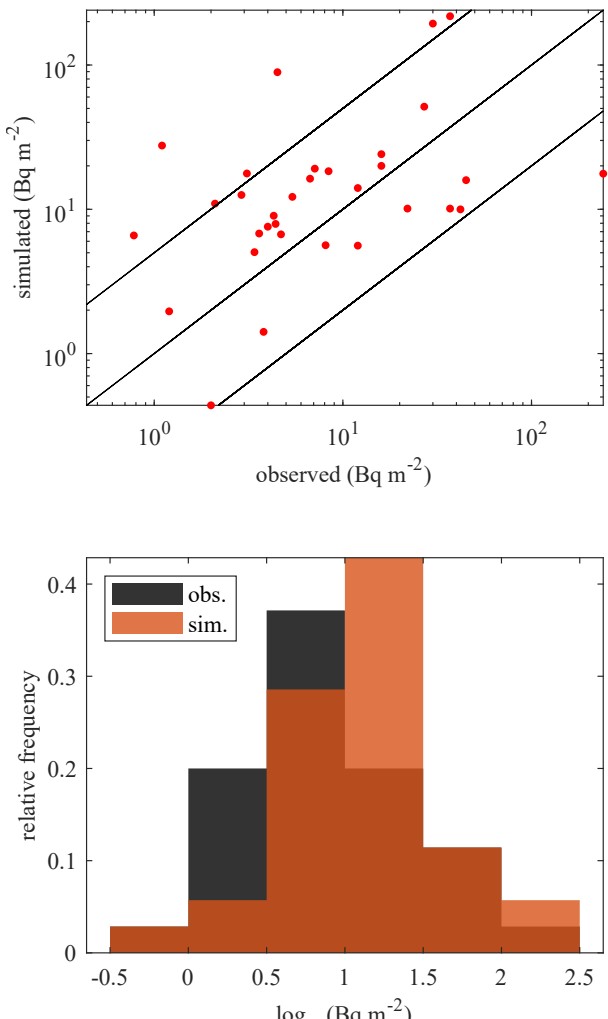

**Figure 9.** Wet deposition of $^{137}$Cs: reference simulation vs. NADP observations.

Table 8. The reference FLEXPART predictions are a better match with the observations in this case compared to the air concentrations as seen previously. Still, a slight bias for overprediction is seen. The optimised FLEPXART simulation is shown in Fig. 10 and in Table 9. A marginal improvement is seen in FB, VG, R and FAC2 and a slight deterioration of MG and NMSE. Overall, we can say that the optimisation process has neither significantly improved or worsened the deposition results.



| FB (0) | MG (1) | NMSE (0) | VG (1) | R (1) | FAC2 (1) |
|--------|--------|----------|--------|-------|----------|
| 0.46 | 0.65 | 7.58 | 6.37 | 0.07 | 0.31 |

**Table 8.** Statistical scores of the reference simulation deposition.

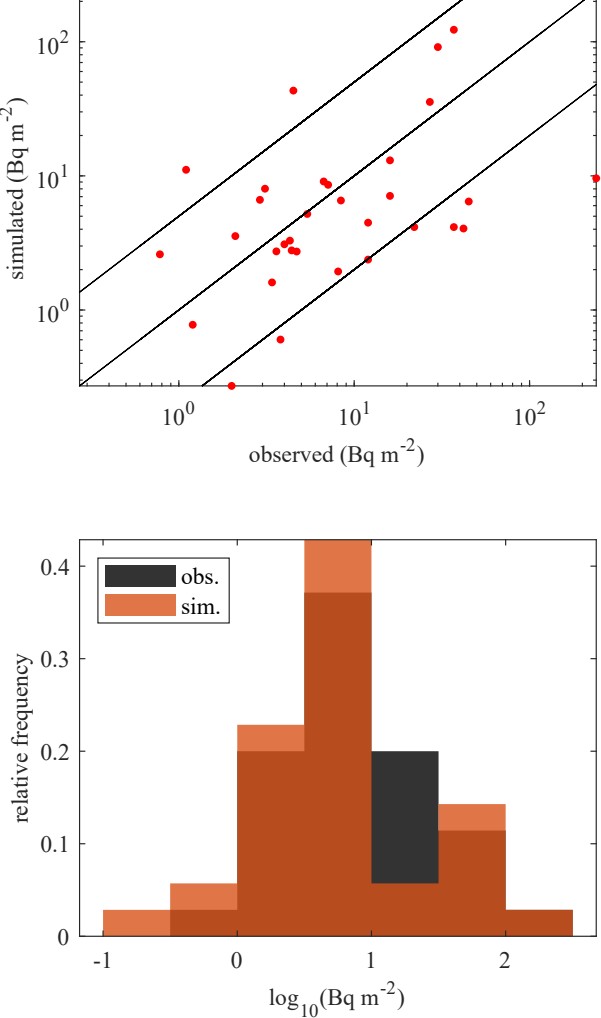

**Figure 10.** Wet deposition of $^{137}$Cs: optimised FLEXPART simulation vs. NADP observations.



| FB | MG | NMSE | VG | R | FAC2 |
|---|---|---|---|---|---|
| (0) | (1) | (0) | (1) | (1) | (1) |
| -0.22 | 1.47 | 8.25 | 6.34 | 0.09 | 0.37 |

**Table 9.** Statistical scores of the deposition from the concentration-optimised FLEXPART simulation.

## 5 Conclusions

Wet deposition plays a crucial role in many atmospheric aspects, such as the transport and diffusion of airborne radionuclides. Simulating wet deposition in atmospheric transport models, however, remains a challenge. Therefore we have developed a post-processing scheme to optimise the wet scavenging rate in ATMs. This methodology was applied to a case study of aerosol-attached $^{137}$Cs following the Fukushima nuclear accident, with the use of the atmospheric transport model FLEXPART. In order to utilise the new optimisation scheme, we accurately determined the partitioning of the scavenging processes as calculated by FLEXPART: (1) below-cloud scavenging by rain, (2) below-cloud scavenging by snow, (3) in-cloud scavenging by cloud condensation nucleation and (4) in-cloud scavenging by ice nucleation. We found that the majority of radionuclides are scavenged below-cloud. The optimisation scheme is able to reduce the originally over-predicted air concentrations of $^{137}$Cs from an average factor 5 to a factor 2. A proposal is made to translate the post-process optimisation results to the physical scavenging coefficients $\Lambda_i$ of the different processes. Using these $\Lambda_i$'s, a FLEXPART simulation could be performed which closely matched the post-process-optimised concentrations. The deposition results of this optimised simulation were compared with measurements made in the U.S. (as an independent check) and show neither a significant improvement nor worsening of the already reasonable prior agreement.

Although applied to FLEXPART, the proposed methodology herein could be applied to any ATM with different scavenging processes modelled. In general, the optimisation scheme will result in a scaling factor of the scavenging coefficient for each scavenging process. In this way, one can circumvent the need to explore the parameter-space of scavenging rates with many separate ATM simulations.

*Code availability.* FLEXPART v10.4: https://www.flexpart.eu/. MATLAB (2021b) code for the analysis: https://doi.org/10.5281/zenodo.7789039

*Author contributions.* SVL: methodology, formal analysis, writing – original draft. PDM: methodology, conceptualisation, writing – review and editing; JC: conceptualisation, supervision. PT: conceptualisation, writing – review and editing, supervision. AD: conceptualisation, software and meteodata access, computing resources, writing – review and editing.



*Competing interests.* The authors declare that they have no conflict of interest.

*Acknowledgements.* The work herein was made possible by using data from the Global Precipitation Climatology Project, the European Centre for Medium-Range Weather Forecasts, the Comprehensive Nuclear Test-Ban Treaty Organisation and the National Atmospheric Deposition Program.



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
