# Peer review of "An optimisation method to improve modelling of wet deposition in atmospheric transport models: applied to FLEXPART v10.4"

_EGUsphere, 2023_

## Author Response (AR1)

**Response to referee comments 1**

1. *It seems that by adopting the optimization parameters xi, the pollutants scavenged by the multiple processes at every timestep would be post-processed to make the atmospheric concentration left in the atmosphere closer to the observation. Could you explain how to consider the influence of the changed atmospheric concentration at one timestep to the concentration at the following one?*

Flexpart calculates deposition at each timestep. However, our method doesn't scale the deposition at each timestep independently. Instead, the deposition is scaled once for all locations and timesteps. The influence of the depletion on the concentrations from one timestep to the next is embedded in the factors $\lambda_i(t)$ of Eq. (14). As an aside, Flexpart does not model any nonlinear effects on air concentration (such as chemical processes).

2. *It would be clearer to explain the method mentioned in section 4.4 and the independent verification setup in section 2.*

Thank you for this suggestion, we will adjust our manuscript accordingly.

3. *a Sensitivity study could be done in independent verification, with few measurements adopted to identify the xi and more measurements left for verification.*

Following this suggestion, we have performed a sensitivity test by using a random subset of the data. Using only 50% of the data provides consistent optimisation parameters that vary only by around 5% compared to usage of the whole dataset. Using the subset-optimised parameters on the remaining data results in a similar variation of the statistical scores. We will update our manuscript with this extra information.

4. *The concentration is a 3D parameter, so how could this paper relate the observation with the delta c in the model? Is the c0 or delta c vertically integrated or are they the value on the ground level?*

The concentrations c0 and $\Delta c$ are not vertically integrated, but represent values as simulated for ground stations at certain locations (see Fig. 1 in the manuscript). These values are calculated during the simulation by the use of 'receptors' in Flexpart. This calculation involves the use of a parabolic kernel, which also extends in the vertical direction (to maximally 150m above ground). More information about the parabolic kernel can be found in Section 8 of 'Stohl et al. 2005: FLEXPART description'.

*Furthermore, The delta c contributed by the in-cloud scavenging is certainly lower than that contributed by below-cloud scavenging at the ground level, leading to the dominant role of below-cloud scavenging.*

The dominant role of below-cloud scavenging, which we observe from the simulations, could indeed be explained by the fact that we consider ground level concentrations. Though, of course, the concentrations as detected by a measurement station may contain particles that have passed through a significant amount of in-cloud scavenging regimes on their way from the release point to the station.

5. *Why do this study choose the interior algorithm to solve Eq. 15? And how many data are actually used to solve the equation?*

We have no specific reason to choose the interior-point algorithm, other than that it is the default algorithm for the optimisation function 'fmincon' in MATLAB. Other algorithms such as a trust-region-reflective algorithm were also applied and showed nearly identical results. The equation was solved for all nonzero measurements available, totalling to 248 data-points across 20 locations covering the northern hemisphere (see Figure 1). This will be clarified in our manuscript.

**Response to referee comments 2**

Main comments

• **Met data**

*In Stohl et al (2011), FLEXPART simulations using ECMWF were compared with those using GFS met data, and quite large differences were found. "The agreement of model results (both using a priori and a posteriori emissions) with measurement data was better with GFS data than with ECMWF data. The fact that this was also found for 133Xe which is not affected by wet scavenging, shows that GFS-FLEXPART captured the general trans- port better than ECMWF-FLEXPART. Furthermore, the wet scavenging of 137Cs was much stronger with ECMWF data than with the GFS data, causing a strong underestimation of 137Cs concentrations at sites in North America and Europe".*

*o     I believe that the location of a low-pressure system northeast of Japan was the main contributing factor in the difference between ECMWF and GFS for this case. Appreciating that repeating the current study using GFS data is a large task, I would encourage this if possible, or at the minimum include some discussion of the abovementioned findings from the Stohl paper. Your Figure 5 shows 1-month averaged ECMWF precipitation; however, I believe this will smooth out local differences like the position of a low-pressure system over 2-3 days which might play a significant role for scavenging.*

*o     Also, the Stohl study found an underestimation of modelled air concentrations compared to measurements, while the current study shows an overestimating, albeit using different FLEXPART model versions. Some discussion of this in the current paper would be appreciated.*

1. We acknowledge the potential impact of the numerical weather model on our results. The optimisation is also not only sensitive to the precipitation, but also the cloud water fields, as this quantifies below and in-cloud scavenging. This information is read from ECMWF and can also differ between NWP's. So our optimisation is also influenced by the choice of NWP in this regard. However, we noticed the substantial impact on the air concentrations when comparing Flexpart v9 vs. v10, using the same NWP data (ECMWF). With the main relevant difference between these versions being the wet deposition scheme, this is where we focused our attention.

2. Using Flexpart v9, which is also used in Stohl (2011), we find an underestimation of the air concentrations as well. Since scavenging is generally decreased for FPv10 compared to v9, the former will result in greater air concentrations. A discussion on the results of the Stohl et al. 2011 study will be added to the manuscript.

• **Aerosol lifetime**

*In Grythe et al (2017) the aerosol lifetime is calculated, using the Fukushima case study and Cs-137 model and observation data. The modelled aerosol lifetime (~10 days) was found to be too short compared to measurements (~14 days), and comparison between FLEXPART v9 (~6 days) and v10 (~10 days) showed that the new wet deposition in FLEXPART perform better than the old scheme (See Fig 5 in Grythe paper). However, a too short aerosol lifetime indicate too quick removal in the model of which scavenging is a significant contribution. This contradicts the results in this current study which found the air concentrations too high compared to measurements and the scavenging needed to be increased. Some comparison and discussion of the current paper and the aerosol lifetime study would be good to include in the paper.*

This contradiction can likely be explained by the fact we use only one particle size bin (0.65 µm) while Grythe et al. 2017 uses multiple bins ranging from 0.4 µm to 6.2 µm. Table 3 in Grythe (2017) provides results for a release with all mass in the 0.65 µm bin, which shows a relative concentration bias of 11. This corresponds to an overestimation of the air concentrations, similar to the overestimation that we found. A discussion with a comparison to Grythe et al. 2017 will be added to the manuscript.

The motivation for our choice of particle diameter is that it corresponds to the main mode found in measurements following the Fukushima accident. We have chosen not to fine tune the particle diameter distribution further, since we observed the dry deposition to be more sensitive to the particle size compared to (below-cloud) scavenging. Our focus was on the wet scavenging due to the differences seen between Flexpart v9 and v10's wet deposition scheme.

• **Use of measurements**

*Including observation error, i.e., uncertainty in the measurements, is key when comparing model and measurements. Do you account for measurement errors? As FLEXPART does not simulate*

*the background variability in the caesium concentrations this should be accounted for when comparing to the measurements, do you subtract a background value from the measurements?*

We didn't take into account measurement error explicitly, as we found this to be negligible compared to the measured values. The caesium background is not taken into account for similar reasons. This background is much smaller than the measured values, usually being on the order of 0.1-1 µBq/m³. If present, it should manifest as a bias at lower concentration values, which is not seen in the data (see Figure 2). The xenon background, on the other hand, is visible in the data (Figure 3) and is removed with a cut-off for the analysis. This information will be added to our manuscript.

• **Deposition vs air concentrations**

*Your study finds that both air concentrations and deposition are overpredicted by the reference simulation (Fig 2 and Fig 9). I would expect deposition to be underpredicted if air concentrations are overpredicted. Does this reflect measurement uncertainty? Please comment.*

There are two competing effects, especially manifesting at longer ranges. One is that an increase in deposition reduces air concentration, as mentioned. Another is that a higher air concentration increases scavenging and thus deposition. These two effects almost cancel out in the reference simulation, leading to a small overprediction of the deposition. The deposition measurements are taken from the U.S. which is some thousands of km's from Fukushima.

• **Case sensitivity**

*Are the scavenging coefficients you derive likely to be case specific to the Fukushima case, and therefore not easily generalisable? The Grythe et al 2017 paper indicate they are case dependent. The abstract and conclusion would benefit from a statement whether the derived coefficients are for Fukushima only or suggested for more 'general' use.*

Thank you for this suggestion. We hypothesise that the scavenging coefficients should be valid more generally, under similar conditions. The NWP, for example, influences the quantification of below- and in-cloud scavenging. Thusly, the optimised coefficients are not necessarily applicable to other NWP's. Similarly, according to Grythe et al. 2017, the scavenging coefficients may depend on a case-by-case basis. We will update our manuscript accordingly.

**List of relevant changes in manuscript**

- Line 9-14: Added references in first two paragraphs of the introduction.
- Line 187-197: Moved the 'translation of optimisation parameters to Flexpart input parameters' section from the results section to methods.
- Line 241-242: Added a discussion on the Cs-137 background.
- Line 280-282: Added a discussion with comparison to Stohl et al. (2012).
- Line 283-296: Added a discussion with comparison to Grythe et al. (2017).
- Line 297-301: Added a discussion on the impact of the choice of numerical weather data.
- Line 349-358: Added a sensitivity analysis of the optimisation parameters.
- Line 379-381: Expanded the discussion of the wet deposition comparison with observations.
- Line 370, 396: Added comments on the applicability of the new parameters to other cases than Fukushima.